# Uptake of community antiretroviral group delivery models for persons living with HIV in Arua district, Uganda: A parallel convergent mixed methods study

**Wani Muzeyi** [1]*, **Semeere Aggrey**[2], **Dennis Kalibbala**[1], **Thomas Katairo** [1], **Fred C. Semitala** [3], **Achilles Katamba**[1], **Irene Ayakaka**[4], **Nelson Kalema**[2]

**1** Clinical Epidemiology Unit, School of Medicine, College of Health Sciences, Makerere University, Kampala, Uganda, **2** Infectious Diseases Institute, College of Health Sciences, Makerere University, Kampala, Uganda, **3** Department of Medicine, College of Health Sciences, Makerere University, Kampala, Uganda, **4** Uganda Tuberculosis Implementation Research Consortium, College of Health Sciences, Makerere University, Kampala, Uganda

* wanixxl@gmail.com

**Data Availability Statement:** The datasets used for the study are available as supplementary attachments.

## Abstract

Community antiretroviral groups (CAGs) is one of the innovative and efficient differentiated service delivery models (DSDM) for reaching persons needing human immunodeficiency virus (HIV) treatment in the community. Since DSDM adoption in Uganda, evidence suggests better care outcomes for patients in DSDM compared to counterparts in routine health facility care. However, uptake of CAG models for eligible community groups of persons living with HIV (PLHIV) has been slow in Arua district, Uganda and stakeholders' perceptions regarding its implementation unexplored. The objective of the study was to determine the uptake, barriers and facilitators influencing CAG model implementation in Arua district, Uganda. We conducted a parallel convergent mixed-methods study from March 2020 to December 2020 at Adumi health centre IV and Kuluva hospital in Arua district. We enrolled and extracted data for every fifth virally suppressed participant on antiretroviral therapy (ART) at the two health facilities. Data were analysed using STATA 13.0. Uptake was determined as the proportion of eligible PLHIV that were enrolled into a group. We performed logistic regression to determine factors associated with uptake. We conducted one focus group discussion per facility among healthcare workers involved in the management of PLHIV. We also conducted 7 focus group discussions among PLHIV across the two facilities. Thematic analysis was used to describe the data. A total of 399 PLHIV were eligible for CAG, 61.6% were female, and 44.9% were on dolutegravir (DTG) based regimen. Uptake was 6.8%, 95% CI (4.7–9.7) and was found to be significantly associated with being divorced or separated in a marriage (OR; 0.14, 95%CI; 0.02–0.92, P = 0.014). Members picking drugs in turns, comforting and encouraging others to take the drugs, and health workers advising them to join and stay with other group members were perceived as facilitators to uptake of community antiretroviral group delivery model. Having few and distant eligible members in the local area to form a group, lack of transport among the member to pick the drugs when it's their turn, misunderstandings and lack of confidentiality amongst the

**Funding:** The authors received no specific funding for this work.

**Competing interests:** The authors have declared that no competing interests exist.

members, and lack of partner disclosure were perceived as barriers to uptake of community antiretroviral group delivery model. Uptake of community antiretroviral group delivery model in Arua district is very low. There may be a need to support community antiretroviral group delivery models with income- generating activities, transport facilitation, closer community drug pick-up points and improved partner disclosure support mechanisms among married group members.

## Introduction

Globally, over 36.9 million people are living with the human immunodeficiency virus (HIV), 59% of whom were receiving lifelong antiretroviral therapy (ART). In 2017 eastern and southern Africa carried 53% of the global HIV burden. In Uganda had about 1.4 million people are living with HIV (PLHIV) of which 73% are enrolled on lifelong ART which is short of the 95% target [1].

Achieving the UNAIDS 95-95-95 goal requires the adoption of innovative and efficient ways of delivering HIV prevention, care and treatment services that address the needs of different groups of clients under care [2].

In November 2015, the WHO launched test and treat and recommended differentiated HIV care and treatment models which includes modifications of client flow, schedules and location of services to adjust to the different needs of groups of clients under HIV care which in turn improves access, coverage and quality of care and thus an increase in the number of people accessing HIV treatment services [3].

In response to the health workforce shortage and the increased demand for HIV treatment following the adoption of test and treat, the Uganda government developed and pilot tested community-based ART delivery and task-shifting models. This included the use of community distribution points and mobile units or mixed models of community-based and facility-based service provision to bring HIV care and treatment closer to the community. These models are critical in removing barriers to accessing ART care and reducing associated cost [4].

Following the success of the models in a few pilot districts, ministry of health Uganda (MOH) rolled out policy guidelines on how to implement these differentiated care models including community ART groups (CAGs) across the country.

The CAG model presents an opportunity for stable clients to access care in the same community where they live through the creation of client groups that provide peer support and alternate picking up of drug refills for the entire group from the facility, all members are encouraged to attend the HIV clinic in person at least once every 6 months at which they get clinical assessment [2].

The benefits of CAGs have been highlighted by several studies, in one study, a retrospective cohort study on four-year retention and risk factors for attrition among members of CAGs in Mozambique reported mortality and LFTU rates among 5729 CAG members were significantly low at 2.1and 0.1 per 100 person-years [5].

In another study on the effect of CAGs on retention in care among patients on ART in Tete province of Mozambique, 12-month and 24-month retention in care from the time of eligibility were 89.5% and 82.3% among patients in the individual care and 99.1% and 97.5% among those in CAGs respectively, CAGs had a greater than fivefold reduction in risk of dying or being LTFU [6].

CAG models also offer benefits to the facility and health sector, a study on the financial impact of DCMs found that an estimated 17.5% could be saved from 2016 to 2020 from implementing the age and stability DCM and four-criteria DCM, respectively, An estimated 46.4% fewer health workers were needed in 2020 for the age and stability DCM compared with undifferentiated [7].

The evidence is clear that clients in CAGs have better retention, low LTFU, low mortality, reduced frequency of clinic visits, cost-effectiveness and low health worker force requirement compared to individual care.

HCW benefits include a reduced workload. Additionally peer support is perceived as an added value of the groups allowing not only sharing of the logistical constraints of drugs refills but also enhanced emotional support [8].

It is envisioned that CAGs and other differentiated ART models will improve the quality of services offered to clients, maximize the efficiency and cost-effectiveness of the country ART program resulting in reduced clinic visits and decreased systems and client challenges [9].

Despite the anticipated benefits of CAG, its uptake among stable HIV positive patients has been low in Arua district. Since its implementation, only 3 groups of 6 people each have been established in Adumi health centre IV and with similar trends of implementation in other facilities, this presents a huge practice gap in the implementation of CAGs in Arua district.

The purpose of this study was to determine the uptake, barriers and facilitators influencing CAG model implementation in Arua district, Uganda.

## Materials and methods

### Study design

We conducted a parallel convergent mixed methods study. We collected and analysed quantitative and qualitative data at the same time, kept the data analyses independent and reported both qualitative and quantitative findings. This was intended to provide a deeper understanding of the implementation of CAGs in Arua district.

### Study setting

The study was conducted in Arua district. The district lies 535 North West of the capital, Kampala with a population of over 500000 and an HIV prevalence of about 3.2%. The district has 1 hospital, two HC IVs and eleven HC III. Study participants were recruited from Adumi health centre IV and Kuluva hospital. Adumi had about 500 PLHIV and Kuluva hospital had about 5000 PLHIV. The two facilities were purposively selected in the district due to the high number of PLHIV that receive treatment from them.

Kuluva hospital is a private for-profit hospital that is run by the church of Uganda while Adumi health centre is fully run by the government and all services are free of charge. Both facilities offer a wide range of services including outpatient, inpatient, maternity, HIV/TB services among others. At Kuluva hospital patients are charged a user fee but all HIV services are free for charge. Each facility runs an ART clinic twice a week.

### Study population

The study was conducted among adult(18 years and above) patients receiving ART for at least 12 months, without treatment regimen change for the same period, with a suppressed viral load, adherence on ART >95%, not pregnant or lactating, on first or 2nd line regimen and with no concurrent illness.

### Data collection

**Quantitative data.** We used systematic sampling where social demographic and medical characteristics of all eligible every 5th participant from the ART register were extracted using a pretested structured data collection tool. Key medical variables included disclosure which referred to PLHIV and their sexual partners being aware of their HIV status, ART regimen (combination of ARVs being taken by an individual) and model of collection of ART from the facility in addition to social demographic variables.

**Qualitative data.** Health workers from these two facilities were purposively selected to take part in the one FGD for HIV care and treatment service providers per facility. The health care provider FGDs both had 12 participants including the clinic incharge, data clerks, social workers, expert clients, nurses, midwives, laboratory personnel, clinicians among other staffs.

Among PLHIV from the two facilities, we conducted at total of 7 FGDs. The FGDs were stratified by sex and whether or not one belonged to a CAG. The participants for the FGD were randomly selected on the day of the HIV clinic. Each FGD had 6–10 participants. FDG were conducted by an experienced social scientist in vernacular (Lugbara) at the respective facility and the PI took field notes. Interviews were audiotaped using a tape recorder. FGDs also enabled the sharing of information on experiences both among the participants and between the healthcare workers.

### Data management and analysis

**Quantitative data.** Data was entered into Epidata and exported to Stata 13 for analysis. Participant's characteristics were summarized as frequencies and percentages. Uptake was determined by dividing the number of clients enrolled in CAG across the two facilities divided by the total number of eligible clients in the study and this was expressed as a percentage. We performed logistic regression to determine factors associated with the uptake of community ART groups. Covariates for the multivariate model were selected a prior from literature. The level of significance of the multivariate model was 0.05.

**Qualitative data.** Recorded interviews were transcribed verbatim and then non-English interview responses were translated into English. Careful attention was given to the local vocabulary used to describe phenomena. Thematic analysis was used to describe the data since this approach is flexible and allows the researcher to describe the data without sacrificing complexity [10].

The following steps were followed: The researchers read data to become familiar (familiarization process) with the discussions, while paying close attention to the patterns that occurred. Codes were generated through data reduction and pattern documentation where the data will be labeled to create categories. Data was coded using open code. The themes were searched from the developed codes and combining codes into overlapping themes to come up with the main themes.

### Ethics approval and consent to participate

Ethical approval to conduct the study was obtained from the Uganda cancer institute research and ethics committee (ucirec@uci.or.ug). Ethical approval was granted number UCIREC REF-13-2019. Written informed consent was obtained from all participants prior to participation in the qualitative study. A waiver of consent was granted for the analysis of quantitative data.

# Results

## Quantitative

Total of 399 participants were enrolled into the study. Of these, majority (74.7%) were from Kuluva hospital.

## Social demographic characteristics

Majority (85.7%) of the participants were over 30 years old. Over 60% were female. The results are summarized in Table 1 below.

## Medical characteristics of study participants

Most (97.2) of the participants had not disclosed their HIV status. The most common (44.9%) regimen was TDF/3TC/DTG. Majority (97%) of the participants were on first line medication. Twenty one percent of the participants were under fast track refills and another 19 percent were taking multi month refills. The results are summarized in Table 2 below.

## Uptake of community ART groups

Only 6.8% (27/399), 95% CI (4.7–9.7) of the eligible participants were enrolled into a CAG. This is illustrated by Fig 1.

**Table 1. Participants social demographic characteristics.**

| Variable | Frequency(n = 399) | Percent (%) |
|---|---|---|
| **Age** | | |
| ≤20 | 9 | 2.3 |
| 20–30 | 48 | 12 |
| >30 | 342 | 85.7 |
| **Sex** | | |
| Male | 153 | 38.4 |
| Female | 245 | 61.6 |
| **Marital status** | | |
| Single | 29 | 7.3 |
| Married | 258 | 64.8 |
| Divorced/separated | 111 | 27.9 |
| **Education** | | |
| None | 92 | 23.1 |
| Primary | 214 | 53.6 |
| Secondary | 71 | 17.8 |
| Tertiary | 22 | 5.5 |
| **Occupation** | | |
| None | 50 | 12.5 |
| Farmer | 253 | 63.4 |
| self employed | 29 | 7.3 |
| Formally employed | 66 | 16.5 |
| **Distance** | | |
| ≤10 | 339 | 85 |
| 11–50 | 53 | 13.3 |
| >50 | 7 | 1.7 |

**Table 2. Participant's medical characteristics.**

| Variable | Frequency(n = 399) | Percent (%) |
|---|---|---|
| **Disclosure** | | |
| Yes | 12 | 3.0 |
| No | 387 | 97 |
| **ART regimen** | | |
| TDF/3TC/DTG | 179 | 44.9 |
| TDF/3TC/EFV | 176 | 44.1 |
| TDF/3TC/NEV | 27 | 6.8 |
| second line | 11 | 2.8 |
| Others | 6 | 1.5 |
| **Regimen type** | | |
| First-line | 387 | 97.0 |
| second line | 12 | 3.0 |
| **Fast track refill** | | |
| Yes | 85 | 21.3 |
| No | 313 | 78.5 |
| **Multi-month refills** | | |
| Yes | 78 | 19.5 |
| No | 320 | 80.5 |

## Factors associated with uptake of community ART groups

Marital status (being separated) was found to be significantly associated with uptake of community ART groups. The results are summarized in Table 3 below.

## Qualitative

This section presents the perceptions that study participants had about community ART groups: The data were categorized into themes that respondents perceived to be facilitators or barriers to CAG model uptake and recommendations for improved uptake.

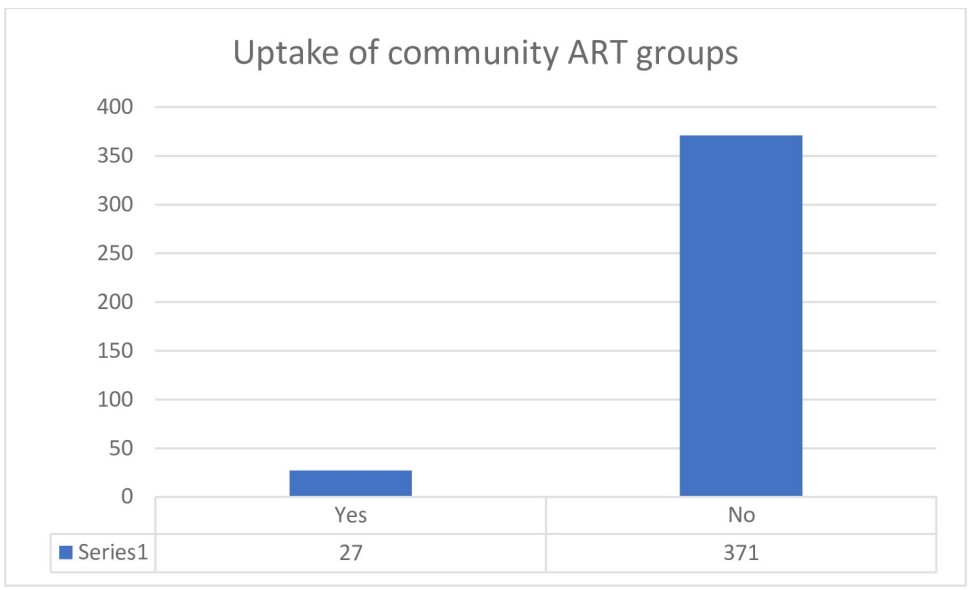

**Fig 1. Bar graph showing uptake of community ART groups in Arua district.**

**Table 3. Factors associated with uptake of community ART groups.**

| Variable | Crude OR | Adjusted OR | 95% CI | P value |
|---|---|---|---|---|
| **Marital Status** | | | | |
| Single | 1.00 | 1.00 | | |
| Married | 0.81 | 0.65 | 0.17–2.50 | 0.535 |
| Separated | 0.16 | 0.14 | 0.02–0.92 | **0.041** |
| **Sex** | | | | |
| Male | 1.00 | 1.00 | | |
| Female | 0.89 | 1.34 | 0.49–2.64 | 0.764 |
| **Education** | | | | |
| None | 1.00 | 1.00 | | |
| Primary | 2.89 | 2.15 | 0.60–7.81 | 0.244 |
| Secondary | 1.77 | 1.94 | 0.39–9.65 | 0.421 |
| Tertiary | 1.48 | 8.87 | 0.39–202.2 | 0.171 |
| **Occupation** | | | | |
| None | 1.00 | 1.00 | | |
| Subsistence farmer | 5.37 | 4.62 | 0.56–37.54 | 0.153 |
| Formal employment | 0.75 | 0.34 | 0.01–9.82 | 0.526 |

## Participant's characteristics

The FGD for the health care workers from each facility was composed of nurses, clinical officers, midwives, data clerks, counsellors, VHTs and expert clients.

Among the PLHIV, FGDs were stratified by Sex and whether or not a participant is a member of a community ART group. The median age among female participants was 41 interquartile range (iqr) 16 while the median age for males was 52 iqr (15).

## Perceived facilitators to uptake of community ART groups

**Members picking drugs in turns.** The respondents noted that picking drugs in turns allows other members of the group to rest and sometimes it gives them time to do other activities like gardening.

*"Other group member pick drugs for you, so you have enough time to rest"* **said a male CAG member from Kuluva.**

**"***It gives you time to focus on other activities like gardening",* **said a female CAG from Adumi health center.**

## Comfort and encouragement to members to take the drugs

The respondents shared that the groups provide comfort and encouragement to members to take their drugs especially the new group members who often feel hopeless and find ART initiation difficult and challenging.

*"When I joined the group I got encouragement from the group members and this comforted me so much and I felt free in my heart now".* **55 year old female CAG member from Kuluva hospital.**

*"When I came alone for treatment, I found it difficult but the group made me feel better, we encourage each other to take the drugs".* **45 year old female CAG member from Adumi health centre**

*"I had lost hope and I regarded myself as a dead and useless person, when I joined the group, the group members comforted me"*, **38 year old male from Adumi**.

### Health worker's advice to join and stay with other group members

Respondents also mentioned that the health worker advises eligible members to create groups and stay with other group members. Being a member of a community ART group allows members to spend less time at the facility since are served faster.

*"I used to pick my drugs alone, was advised not to stay alone but stay with other group members. You spend less time at the health center, because you're served faster".* **Said a male CAG member from Adumi and a female CAG member from Kuluva hospital.**

*"The patients are few and therefore we serve them faster, this reduces on our work load"*, **said a health worker from Kuluva hospital**.

### Perceived barriers to uptake of community ART groups

**Few and distant eligible members in the local area to form a group.** Respondents also reported that some clients are too far apart which makes it hard to start a community ART group in the local area.

*"Some people are far while others are near, for those that are far it's difficult to start a group".* **64 year old female from Kuluva**.

*"In Barize village, we are few and distant so that makes it hard for us to form a group."* **38 year old female non CAG member from Adumi**

**Lack of transport among the member to pick the drugs when it's their turn.** They also mentioned that there are situations where members lack transport collect drugs when it their turn which is very challenging.

*"Sometimes I would miss to pick drugs for me other members because I would not have transport, this forced to leave the group".* **47 year old female, non CAG from Adumi health center IV.**

### Misunderstandings among group members

They also reported that the experience misunderstandings among group members and to the worse, some members lose interest and end up leaving the group.

*"Not understanding one another in the group makes some people lose interest and end up leaving the group"*, **said a health worker from Adumi health center.**

**Lack of confidentiality amongst the members.** They also reported that some group members inform the village members about the status of other members in the ART groups which is very challenging.

*"Some group members tell everyone in the village about the groups and who picks drugs for you, this makes some people to leave the groups because of their secrets being leaked by group members"*, **said a male participant from Kuluva hospital.**

## Lack of partner disclosure and involvement

The respondents reported lack of partner disclosure amongst the married members of the group as another challenge to uptake of community ART groups. Eligible married members in the community refuse to join the groups because they fear their partners may inadvertently learn of their spouse's HIV status. They added that, for those who manage to join, they even hide their medications far from their partners.

> "*Some married people don't want their partners to know their status, so when you advise them to join the group, they refuse*", **says 57 year old female from Kuluva**,

> "*Some women even hide their HIV medications from their partners so it's hard for them to join these groups because their husbands can find out*", **says a 45 year old male non CAG member from Adumi**

**Long waiting time at the facilities when having other medical conditions.** They reported that they experience delays at the health facility when they go to pick the drugs with other medical conditions and this discourages them from coming back to the facility.

> "*When you come to pick drugs and you have other conditions, you're sent to OPD where you wait for hours with other patients who don't have HIV, this makes not to want to come back.*"**43 year old male non CAG member from Kuluva hospital.**

## Discussion

Majority of the participants were from Kuluva hospital and this was because it had the largest number of participants compared to the HC IV. In our study, majority of the participants were on TDF/3TC/DTG, this was because this particular regimen was reserved for stable patients.

In our study, only 6.8% of all eligible participants from the two health facilities had enrolled into a community ART group. This is similar to findings from a process evaluation study on using differentiated care models to achieve national treatment goals conducted in Malawi, which reported a 6% enrolment into community ART groups [11].

Currently the national treatment targets of Uganda have shifted to the 95% targets from 90% targets. This calls for more efforts to be directed towards improving the uptake of community ART groups if the new treatment targets are to be achieved timely.

## Perceived facilitators to uptake of community ART groups

Feeling better, prospects of living longer, family support, information about ART, support for income generating activities, disclosure of HIV status, prayers and transport support were among the facilitators [12].

In our study, being assigned to CAG, spending less time at the health facility, getting comfort, encouragement and counsel from group members were noted as facilitators to uptake of community ART groups.

## Perceived barriers to uptake of community ART groups

Partner involvement was identified as a key facilitating factor to the uptake of community ART groups [13], in our study participants reported that it was difficult for eligible patients who had not disclosed to their partners to join community ART groups. This is because such

patients would want to keep their treatment details from their partners. Measures should be put in place to encourage disclosure and partner involvement among eligible patients. This is consistent with the quantitative finding were separated couples were significantly less likely to join a community ART group compared to the single counterparts. This could because after separation, clients become more discrete with their status. Close attention should be given to this group to encourage disclosure.

Adolescent and youth-specific issues around disclosure as the main barriers to joining community ART groups [14]. It is worth noting that in our study, not even a single youth or adolescent was enrolled into any community ART group. There is need to understand individual- and system- level challenges to the uptake of CAG among adolescents living with HIV.

Lack of confidentiality in the treatment centres and among group members has also been cited a barrier to joining community ART groups [12]. Eligible members need to be educated about the importance of keeping group secrets.

Lack of transport was noted as a barrier towards uptake of community ART group model, this is consistent with findings of a study in Zimbabwe were participants identified the cost of transportation to the health facility as a negative element of their current treatment [15]. This probably signifies that PLHIV in rural areas have to travel significantly long distances in order to access HIV care.

The associated user fees that come with a hospital visit to collect group drugs was also identified as a barrier at Kuluva hospital. This being a PNFP, it levies charges on other services offered in case of other medical conditions, but ART drugs are free. This is consistent with the findings from the Zimbabwe study [15]. For patients at Adumi health centre IV being a purely government facility, all services are free of charge, however, group members noted that in case of other medical conditions are forced into the same waiting line as other patients and this was a barrier.

## Supporting information

**S1 Data. Datasets used in the study.**
(XLS)

## Author Contributions

**Conceptualization:** Wani Muzeyi, Semeere Aggrey.

**Data curation:** Dennis Kalibbala.

**Formal analysis:** Wani Muzeyi, Dennis Kalibbala.

**Investigation:** Thomas Katairo.

**Methodology:** Wani Muzeyi.

**Software:** Wani Muzeyi, Dennis Kalibbala.

**Supervision:** Semeere Aggrey, Fred C. Semitala, Achilles Katamba, Irene Ayakaka, Nelson Kalema.

**Validation:** Nelson Kalema.

**Visualization:** Thomas Katairo.

**Writing – original draft:** Wani Muzeyi, Semeere Aggrey, Dennis Kalibbala, Thomas Katairo, Fred C. Semitala, Achilles Katamba, Irene Ayakaka, Nelson Kalema.

**Writing – review & editing:** Wani Muzeyi, Semeere Aggrey, Dennis Kalibbala, Thomas Katairo, Fred C. Semitala, Achilles Katamba, Irene Ayakaka, Nelson Kalema.

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
