## [Decision Letter · Decision Letter 0]

9 Aug 2022

PGPH-D-22-00085

Uptake of community antiretroviral group delivery models for persons living with HIV in Arua district, Uganda: a parallel convergent mixed methods study

Dear Dr. Muzeyi,

Thank you for submitting your manuscript to PLOS Global Public Health. After careful consideration, we feel that it has merit but does not fully meet PLOS Global Public Health’s publication criteria as it currently stands. Therefore, we invite you to submit a revised version of the manuscript that addresses the points raised during the review process.

We look forward to receiving your revised manuscript.

Kind regards,

Rebecca M Flueckiger, Ph.D.

Academic Editor

Journal Requirements:

1. We suggest you thoroughly copyedit your manuscript for language usage, spelling, and grammar. If you do not know anyone who can help you do this, you may wish to consider employing a professional scientific editing service.

2. Please provide additional details regarding participant consent. In the ethics statement in the Methods and online submission information, please ensure that you have specified whether consent was written or verbal/oral. If consent was verbal/oral, please specify: 1) whether the ethics committee approved the verbal/oral consent procedure, 2) why written consent could not be obtained, and 3) how verbal/oral consent was recorded. If your study included minors, please state whether you obtained consent from parents or guardians in these cases. If the need for consent was waived by the ethics committee, please include this information.

3. In the online submission form, you indicated that "The datasets used and/or analyzed during the current study are available from the corresponding author on reasonable request.". All PLOS journals now require all data underlying the findings described in their manuscript to be freely available to other researchers, either 1. In a public repository, 2. Within the manuscript itself, or 3. Uploaded as supplementary information.

5. We ask that a manuscript source file is provided at Revision. Please upload your manuscript file as a .doc, .docx, .rtf or .tex with PDF

6. Please provide separate figure files in .tif or .eps format and remove the embedded figure from the manuscript file.

Additional Editor Comments (if provided):

Reviewers' comments:

Reviewer's Responses to Questions

**Comments to the Author**

1. Does this manuscript meet PLOS Global Public Health’s publication criteria? Is the manuscript technically sound, and do the data support the conclusions? The manuscript must describe methodologically and ethically rigorous research with conclusions that are appropriately drawn based on the data presented.

Reviewer #1: Partly

Reviewer #2: Partly

2. Has the statistical analysis been performed appropriately and rigorously?

Reviewer #1: Yes

Reviewer #2: No

3. Have the authors made all data underlying the findings in their manuscript fully available (please refer to the Data Availability Statement at the start of the manuscript PDF file)?

Reviewer #1: No

Reviewer #2: No

4. Is the manuscript presented in an intelligible fashion and written in standard English?

Reviewer #1: Yes

Reviewer #2: Yes

5. Review Comments to the Author

Reviewer #1: This study assesses the uptake of the community antiretroviral therapy group (CAG) treatment models for persons living with HIV (PLHIV) in Arua, Uganda, and identifies facilitators and barriers to patient participation. The authors used a convergent-parallel mixed method design collecting data for quantitative and qualitative analysis from patient registers and stakeholder focus group discussions, respectively. The findings produced will provide valuable insight into the factors impacting successful implementation of the CAG treatment model in similar settings.

However, there are several issues throughout the paper that do not meet the publication criteria which need to be addressed before I can recommend acceptance.

Data Availability Guidelines

1. The statement included in the submission says the data can be made available upon “reasonable request,” but there is no clear definition for what those entails. The journal requires the Data Availability Statement submitted by authors to include a detailed explanation to justify restrictions to the data. The current statement needs to be revised to describe those details for this dataset.

2. The guidelines also state that it is not acceptable for an author to be the only point of contact (POC) listed. The statement currently only lists the corresponding authors as the POC, but an additional non-author institutional (e.g., a data access committee, ethics committee or other institutional body) POC should also be provided to meet journal requirement to ensure durable access to the data.

General

1. There are minor issues throughout the paper that need to be addressed including typos, use of informal language, and repeated or incomplete sentences. An example is the repeated use of the “picking drugs” instead of “picking up drugs” in the results section.

2. Acronyms should be written out fully when first used in the text. (See methods and results of Abstract and Introduction sections.)

Abstract

• A “Keyword” section is included but it was left blank.

Introduction

• This paper provides a great introduction to the topic of interest; however, I suggest reviewing the section while considering the structure and flow of the paragraphs and revising as needed to improve readability. I noted some redundant sentences in part of the text in this section.

Methods

1. The methods section is well organized and gives a clear overview of the approach; however, it is very brief. It would be helpful to expand the section and more thoroughly describe the steps and tools used to collect the data. This will facilitate replication of the study.

2. An example of key elements to discuss more thoroughly are the criteria used to determine “eligible participants” to include in the quantitative analysis, the structured data extraction tool used to collect data from the patient registers, and the standards used to include participants in the qualitative analysis using a purposive sampling approach.

3. The description of the quantitative analysis also needs to be expanded, particularly logistic regression. The covariates and significance of including them in the regression model were not described.

4. Finally, some sections of text are unclear or appear to be incomplete. (See the last sentence in the paragraph discussing the qualitative sampling procedure and the last bullet of the paragraph discussing data management and analysis of the qualitative data.)

Results

1. The statistical analysis appears to have been appropriately conducted, but as with the methods the section is very brief. I would recommend expanding discussion of the results, particularly the results from the logistic regression. The abstract has a better description of this analysis in the actual results section, and I would strongly recommend reviewing and revising as needed.

2. Table 1 splits across two pages (5-6) thus the portion on page 6 needs to also include the table title followed by the word (continued) and the column headers. Also, the contents of the table review as some variable labels are missing.

3. In Figure 1, the series heading in the legend needs to be defined (it currently appears as “Series1”) or it is best practice removed from the figure and ensure the title is descriptive enough to assist with interpretation.

4. Inclusion of the terms “table” or “figure” the titles for Table 2-3 and Figure 1 is redundant and the titles should be revised to be as descriptive as seen in Table 1.

5. Some quantitative appear to be missing their corresponding symbols.

Discussion

1. The discussion explores the results from the qualitative analysis thoroughly and makes valid interpretation.

2. Interpretation of the quantitative results is brief and should be expanded to highlight the importance of these findings to the study.

3. The connection made between the logistic regression results and the qualitative result regarding marital status and partner involvement does not seem accurate and need clarification. The OR compares each exposure group to the reference group and not necessarily to each other. The current interpretation implies comparison of exposure group to each other instead of the reference group (marital status =single).

Reviewer #2: Overall, this is an important study and focuses on important matters pertaining to HIV prevention and control. My comments are provided to enhance the publication.

My ability to make a sound assessment as a reviewer has been limited due to insufficient information included in the paper. This relates, in particular, to insufficient description of the methods and other important aspects of the study which makes it hard to support or refute the conclusions drawn by the researchers.

Research question

Additionally, another matter that I wish to highlight pertains to the study objective. In the abstract the authors write that the objective is to determine the uptake, barriers and facilitators influencing CAG model implementation in the Arua district. In the body of the paper (under Introduction) they state that the purpose of the paper is to explore the perceptions of key stakeholders regarding the implementation of CAGs. It would be important to align the wording as exploration of perceptions and specifically exploring facilitators and barriers can be argued to be different. In the Results section, the authors state that they report on the ‘perceptions that study participants had about community ART groups…’. They then proceed to only report on facilitators and barriers.

More specific comments

Methods

Study design and analysis

Justification for the use of mixed-methods should be included in the paper. Mixed methods studies implies that the data are ‘mixed’. How and when the data had been mixed during the analytical stage should be made explicit.

Justification for the study types should also be included – for example why focus group discussions as opposed to any other form of qualitative enquiry.

Context

More should be done to make the reader understand the context (e.g. is this a rural area; how many people living in the area; how do the two hospitals related to each other; do they serve different populations, etc)

Eligibility

Defining eligibility would be an important start for the reader. The authors could have described the CAG programme in more detail to ensure that the reader understands exactly what it is, when it runs, what the intent and expectation of government is (e.g. are all patients expected to join these groups), what are the eligibility for joining the group, what is the national average, does uptake differ by rural or less rural areas, etc).

Then the eligibility criteria for selection into the analysis should also be made clear. They speak of adult populations but what is the definition of adult in the country and in the study. Eligibility during data analysis was defined as “number of clients in a CAG divided by the total number of eligible clients…). It is therefore useful to know how eligibility is defined.

The database from which the quantitative data was drawn should also be described in more detail as should the manner in which the authors extracted the data. For example, is the database electronic? Did they review every 5th record manually? They state that all eligible patients were recruited from the facilities but I would imagine that if they did record reviews then the participants would be selected from a list and would not be recruited? Was the analysis disaggregated by hospital and were there any differences?

Qualitative data

For the qualitative data, insufficient information is provided. Who is the sample? Were the names taken from the list on the register? Did they select people from existing CADs (i.e. how were they recruited?) If not, then what guided their selection process (e.g. what were the criteria?). Why were PLHIV and health care workers selected? What is the justification for this? Additional information on selection is listed in the Data Collection section – this should be shifted to the appropriate sub-section. Again, the authors refer to eligible participants, yet eligibility (for both quantitative and qualitative studies) had not been adequately discussed. What is the justification for stratification? They also stratified according to enrolment in the CAG but they do not report on the numbers per group, nor whether there were differences in what these participants reported in the Results section. Finally, how did they notify and select the participants for inclusion in the FGDs?

In the results section, the FGD participants are explained a bit further (and I feel that this should have been justified in the methods section). The sample include a range of healthcare workers including data clerks counsellors, midwives. What is not clear is how many people in total were ultimately included in the qualitative part of the study? How many per group? How and how many according to the stratification criteria? Did they mix the health care workers with the patients?

In terms of general reporting of the results, the analysis does not describe any specific experiences regarding implementation. The ‘themes’ could have been merged or themed into higher-order themes. They seem to very much in parts be extractions or paraphrases of specific phrases in the data. Some of the themes are not described/explained in detail and could benefit from more description based on the data.

It would also be beneficial to get a sense of whether some of the results reported in this section differed according to the hospital that patients attended.

The use of quotations was well done.

Data extraction tool

It would be useful to explain to the reader what the key sections/topics/headings in the data extraction tool were. What were the variables? Was information manually entered into the ‘tool’?

With regards to disclosure of HIV status – it would be useful for the reader to have a definition of what this refers to according to the definition of this data element? In the local context are patients expected to disclose to anyone specific. This will enrich the readers’ understanding of the study results.

Results

Quantitative data

• Could the majority of patients enrolled in the study (i.e. from Kuluva Hospital) be because it is a larger hospital; and if every 5th patient was selected it would be expected that most of the sample would come from this hospital?

• Could there be a more detailed age breakdown of those >30- given that the sample is dealing with adult patients?

• It would be useful to state a bit more of the results in the text of the paper, than refer the reader to the table (e.g. information on marital status is presented in Table 1 but not referred to in the text).

• It is useful to know what the ART regimen is? Could it be explained how this information relates to the study? Would it be expected that patients on certain regimens would be more or less likely to join a CAG?

• One of the most important participant data is reported in Table 2 – fast-track/multi-month refills. This is not mentioned in the text of the results section at all and should be highlighted, and also discussed in the Discussion section.

Discussion

In general the discussion section could be strengthened. More care could be taken to differentiate between findings from the study versus from other papers. In some instances it is not clear whether some referenced sentences refer to the study or the referenced study. Overall, the discussion could benefit from more detailed and explicit ‘mixing’ of the data.

Under the section Perceived barriers to uptake of community ART groups, the second half of the paragraph suggests that separated couples were significantly less likely to join a community ART group. I may have misunderstood it, but this seems to contrast to what is suggested above.

A comment is made regarding adolescent and youth-specific issues. However, this sample is comprised of only adults. Nothing is stated regarding youth or adolescents in the results section. Hence it does not seem appropriate to comment on this if the sample excluded this age group. Thus having a definition of adult and eligibility would be useful.

6. PLOS authors have the option to publish the peer review history of their article (what does this mean?). If published, this will include your full peer review and any attached files.

**Do you want your identity to be public for this peer review?** For information about this choice, including consent withdrawal, please see our Privacy Policy.

Reviewer #1: No

Reviewer #2: No

---

## [Decision Letter · Decision Letter 1]

22 Dec 2022

Uptake of community antiretroviral group delivery models for persons living with HIV in Arua district, Uganda: a parallel convergent mixed methods study

PGPH-D-22-00085R1

Dear Dr. Muzeyi,

We are pleased to inform you that your manuscript 'Uptake of community antiretroviral group delivery models for persons living with HIV in Arua district, Uganda: a parallel convergent mixed methods study' has been provisionally accepted for publication in PLOS Global Public Health.

Best regards,

Henry Zakumumpa, PhD

Academic Editor

We thank the authors for attending to the corrections recommended by our two reviewers. We are delighted to accept this revised manuscript.

Reviewer Comments (if any, and for reference):

Reviewer's Responses to Questions

**Comments to the Author**

1. If the authors have adequately addressed your comments raised in a previous round of review and you feel that this manuscript is now acceptable for publication, you may indicate that here to bypass the “Comments to the Author” section, enter your conflict of interest statement in the “Confidential to Editor” section, and submit your "Accept" recommendation.

Reviewer #3: All comments have been addressed

2. Does this manuscript meet PLOS Global Public Health’s publication criteria? Is the manuscript technically sound, and do the data support the conclusions? The manuscript must describe methodologically and ethically rigorous research with conclusions that are appropriately drawn based on the data presented.

Reviewer #3: Yes

3. Has the statistical analysis been performed appropriately and rigorously?

Reviewer #3: Yes

4. Have the authors made all data underlying the findings in their manuscript fully available (please refer to the Data Availability Statement at the start of the manuscript PDF file)?

Reviewer #3: Yes

5. Is the manuscript presented in an intelligible fashion and written in standard English?

Reviewer #3: Yes

6. Review Comments to the Author

Reviewer #3: Comments have been addressed

7. PLOS authors have the option to publish the peer review history of their article (what does this mean?). If published, this will include your full peer review and any attached files.

**Do you want your identity to be public for this peer review?** For information about this choice, including consent withdrawal, please see our Privacy Policy.

Reviewer #3: **Yes: **Dr Omona Kizito
